# Academic Emotions and Regulation Strategies: Interaction with Higher Education Dropout Ideation

Daniel Enguídanos *, Javier Aroztegui, Manuel Iglesias-Soilán, Irene Sánchez-San-José and Juan Fernández

Departamento de Investigación y Psicología en Educación, Facultad de Psicología, Campus de Somosaguas, Complutense University of Madrid, 28223 Pozuelo de Alarcón, Spain; jarozteg@ucm.es (J.A.); manuelig@ucm.es (M.I.-S.); isanch16@ucm.es (I.S.-S.-J.); jfernandez@psi.ucm.es (J.F.)
* Correspondence: denguida@ucm.es

**Abstract:** (1) Introduction: This article addresses the relationship between students' emotional experiences—mediated by their regulation strategies—and their intention of dropping out. (2) Materials and Methods: An ad hoc questionnaire was designed based on Pekrun's Control-Value Theory of Achievement Emotions focusing on three different settings (study, classroom, and exam). Data were collected from 2183 university students. Descriptive, frequency, and correlation analysis were conducted. Also, linear regression analyses and scatter plots were performed. A comparative regression analysis was used with the aim of facilitating the understanding of the mediating effect of emotional regulation strategies. (3) Results: Academic emotions were found to have a significant impact on dropout ideation. The effects of emotional regulation strategies as significant moderators in this relationship were observed, exhibiting variations depending on the context. (4) Discussion: Dropout ideation escalates in the presence of elevated levels of unpleasant emotions and diminished levels of pleasant emotions. The utility of possessing effective emotional regulation strategies becomes evident in mitigating dropout ideation during emotionally challenging academic situations.

**Keywords:** academic emotions; university; dropout; emotional regulation strategies





## 1. Introduction

University education significantly influence quality of life [1,2]. Graduates experience higher employment opportunities and salaries, both of which increase with their educational attainment. Moreover, university education enhances other valuable skills, such as ICT (Information and Communication Technology) proficiency [3,4].

University education yields notable social and community advantages. Graduates exhibit parenting styles conducive to heightened tolerance, receptivity to diverse cultures, and a heightened sense of global community [4]. Moreover, they frequently engage in community-oriented activities, such as volunteerism and participation in civic organizations [5]. Furthermore, gender disparities in unemployment tend to diminish among individuals with tertiary education qualifications [6].

In acknowledgment of the pivotal role played by higher education in nurturing a nation's resilience and development, the member states of the European Union have established an ambitious objective aimed at guaranteeing that a minimum of 45% of individuals between the ages of 25 and 34 attain a higher education qualification by the year 2030 [7].

This objective aims to equip the younger generation with the requisite skills and knowledge to address the intricate challenges of the future, while fostering innovation and growth across a range of sectors.

Therefore, it comes as no surprise that the Gross Enrollment Ratio (GER) for tertiary education has experienced a substantial increase on a global scale and within the EU member states over the past five decades [8].

However, a substantial proportion of university students drop out before successfully completing their degree programs [9]. In OECD countries, the average attrition rate among bachelor's degree program students is 12% before the commencement of their second year. This rate increases to 21% by the conclusion of the nominal program duration and further escalates to 23% three years later [4]. The abandonment of undergraduate studies not only reduces the prospects of realizing the aforementioned advantages but also signifies a substantial misallocation of effort, final resources, and, sometimes, both public and private investments that yield no return and can even result in lifelong student debts in certain countries [10,11]. Thus, the investigation of the processes and factors contributing to university student dropout becomes of paramount importance.

It is crucial to differentiate between students who voluntarily withdraw from their program to pursue alternative options (such as transferring to another institution or changing majors) and those who drop out due to financial constraints, academic pressure, or other factors [12]. However, this differentiation is not consistently delineated in official reports, potentially complicating the interpretation of the data.

While various models aim to elucidate university dropout, a consensus exists among many of them, agreeing that it results from a multifaceted interplay between individual, socio-economic, and institutional factors [13]. For instance, certain models have identified noteworthy variables, such as gender—with higher dropout rates among males than females—socio-economic status, parental educational attainment, and prior academic performance (with lower dropout rates associated with higher socio-economic status, better-educated parents, and greater prior academic achievements) [14,15].

While some models incorporate cognitive factors, such as prior expectations or cognitive strategies, in their endeavor to elucidate university dropout [16,17], only a limited number regard emotional experiences as a significant variable. However, emotions an intrinsic component of the academic context [18,19]. Research has shown that emotions can have an impact on not only well-being but also on various facets of the learning process, such as engagement, attention, and memory, with potential consequences for academic performance [18,20–22]. This, in turn, may influence the intention to drop out [23]. Positive emotions such as interest, enjoyment, and pride have the potential to bolster engagement and performance, initiating a cycle of even more positive emotions, and so forth. Conversely, negative emotions such as anxiety or boredom can result in disengagement and poor performance, perpetuating a cycle of escalating negative emotions, leading to a downward spiral [24].

Hence, to attain a more comprehensive understanding of the intricacies surrounding dropout phenomena, it is essential to integrate emotional factors into extant models. Subjective variables, such as emotions, beliefs, and attitudes, can wield substantial influence in shaping behavioral intentions, including dropout ideation [25].

The incorporation of emotional factors becomes particularly crucial in instances where students grapple with personal or psychological challenges that may incline them toward contemplating dropout [20]. For instance, a student who is facing homesickness or social anxiety may experience detachment from their academic community and may be more likely to consider leaving their program. By comprehending and addressing the emotional factors involved in such scenarios, universities can provide the essential support and resources required to assist students in surmounting these obstacles and staying on their academic path.

Overall, through the inclusion of emotional factors within current models of student retention, universities can acquire a more intricate comprehension of the factors motivating students to contemplate dropout. Consequently, they can formulate interventions that are not only more efficient but also more precisely tailored to tackle these concerns. This, in turn, can result in enhanced retention rates, heightened student achievement, and a more nurturing and inclusive academic environment.

Therefore, the connection between emotions and dropout ideation is shaped by emotional regulation strategies, which possess the ability to moderate the impact of one on

the other. While a substantial body of literature exists regarding emotional intelligence and emotional regulation [26–29], only a limited number of studies have explicitly explored their moderating influence on dropout ideation or behavior. Nonetheless, existing models on emotional regulation—such as the ERAS model [30]—have yielded encouraging findings with respect to the capacity of emotional regulation strategies to moderate academic emotions.

The aim of this study is to acquire a deeper understanding of the connection between academic emotions and dropout ideation, while also exploring the mediating role of students' emotional regulation strategies. This contribution seeks to expand upon the limited scientific literature on the subject [30,31]. The utilization of dropout ideation as a surrogate for dropout rates primarily stems from considerations of information accessibility and privacy. However, it is essential to underscore the significance of dropout ideation itself, as proactive interventions should ideally be deployed to forestall dropout. Consequently, the variables antecedent to dropping out emerge as central focal points of interest.

To achieve this, seven academic emotions have been chosen, drawing from previous models that delve into academic emotions [32–34]: anxiety, boredom, frustration, hopelessness, satisfaction, surprise, and enjoyment.

These emotions will be assessed within the context of university-level education, encompassing study, classroom, and examination scenarios (students will be asked about their emotional experiences in one of those three academic situations). They will be analyzed alongside data collected on participants' dropout ideation and their perceived efficacy in employing emotional management strategies.

## 2. Materials and Methods

### 2.1. Participants

A total of 2183 students from all faculties at the Complutense University of Madrid participated in the study, representing 3.2% of the university's overall population. Concerning the participants' gender distribution, there were 1614 females (73.93%), 516 males (23.64%), 25 individuals who identified as "another gender" (1.14%), and 28 respondents who chose the option "I prefer not to disclose" (1.28%). The predominance of female participants can be attributed to two factors: the higher representation of female students at the Complutense University of Madrid (62.5% in 2021), and a lower willingness among students from traditionally male-dominated fields, such as engineering, mathematics, and natural sciences, to take part in the study. In terms of age demographics, there were 726 students under the age of 20 (33.25%), 1106 between 20 and 24 years (50.66%), and 351 students aged 25 or older (16.09%).

In terms of educational attainment, 1919 students were pursuing a bachelor's degree (87.91%), 255 were enrolled in an official master's program (11.68%), 2 were working toward a professional title (0.1%), and 7 respondents chose the option "other" (0.32%). Regarding the academic standing, there were 632 first-year students (28.95%), 455 second-year students (20.84%), 407 third-year students (18.64%), 444 fourth-year students (20.34%), 123 fifth-year students (5.63%), 32 sixth-year students (1.47%), and 90 students who selected the option "not applicable" (4.12%). The fifth- and sixth-year students corresponded to bachelor's degrees that extend beyond the usual four years.

The participants were categorized into three groups, one for each of the examined contexts, according to their month of birth. There were 724 (33.17%) participants in the study context, 752 (34.45%) in the classroom context, and 707 (32.39%) in the exam context.

### 2.2. Assessment Instrument

A quantitative assessment instrument was custom-designed, drawing inspiration from previous tools that align with the adopted approach [32–34]. The assessment instrument is intended to evaluate various distinct areas (as outlined below), with each item carrying independent significance.

The instrument comprises four sociodemographic items, which encompass gender, age, educational level, and academic year.

Dropout ideation was quantified using a single 5-point rating scale item that gauged the frequency of dropout thoughts.

Seven distinct academic emotions were evaluated. Specifically, anxiety, boredom, hopelessness, enjoyment, and surprise were derived from the Achievement Emotions Questionnaire—Short (AEQ-S) [32]. Frustration was derived from the Academic Emotions Scale (AES) [33], while satisfaction was derived from the Short Academic Motivation Scale (SAMS) [34]. Each emotion within each educational context (study, classroom, or exam) was assessed using three 5-point rating scale items that measured the frequency of an emotion's presence.

Here are some examples of emotion assessment items tailored to specific educational contexts:

- Anxiety in the exam context: *Taking an exam makes me feel anxious*.
- Boredom in the classroom context: *In general, I get bored in class*.
- Frustration in the study context: *I get frustrated when I don't understand what I'm studying*.
- Hopelessness in the exam context: *I won't do well on the exams*.
- Satisfaction in the study context: *I feel good about what I achieve when I study*.
- Surprise in the classroom context: *I find new ideas in what we do in class (explanations, activities. . .)*.
- Enjoyment in the study context: *I enjoy studying*.

The assessment included the regulation of both pleasant and unpleasant emotions strategies within each educational context (study, classroom, or exam). The effectiveness of these strategies was evaluated using a 5-point rating scale item to measure their efficacy level.

### 2.3. Procedure

Primarily, the assessment instrument was created and administered using the online platform Microsoft$^{TM}$ Forms$^{TM}$, facilitating its completion and enabling a broader participant reach. This format is not only more sustainable and cost-effective by eliminating the requirement for paper, but it is also promotes anonymity, thereby reducing the potential influence of specific biases that might impact the evaluation, such as social desirability bias [35].

The assessment instrument was subsequently distributed to the student body of the Complutense University of Madrid via an institutional email sent by the students' vice-rector on 17 October 2022. Clear instructions for completing the assessment were provided, along with details about the study's objective, to ensure an ethically sound procedure. Every participant explicitly consented to their informed and voluntary participation, adhering to the regulations stipulated by European data privacy laws [36].

### 2.4. Data Analysis

Regarding the statistical analyses presented in this report, descriptive and frequency analyses were conducted using SPSS v. 26.0.0.0. The correlations were also examined using SPSS. Linear regression analyses and scatter plots were performed using Python libraries (pandas v. 1.5.2, numpy v. 1.24.2, scipy v. 1.10.0, statsmodels v. 0.13.5, and matplotlib v. 3.3.4) running on a Jupyter notebook.

One of the central hypotheses pertains to the extent to which emotional regulation strategies can mitigate dropout ideation associated with emotions. This measurement can be expressed as a ratio: the percentage of dropout ideation attributed to effective regulation strategies in comparison to the level of dropout ideation in the absence of effective regulation strategies. To calculate this, regression lines are utilized for both effective and ineffective regulation strategies:

(a) The increase in dropout ideation attributed to emotions in the absence of effective strategies can be represented as follows in Equation (1) as *edi* (emotion dropout increase), with $f_{12}$ being the emotion and dropout ideation regression line for people with poor emotional regulation strategies (responses to strategy performance were

1 or 2), $f_{12}(1)$ the regression line value at a low emotional level, and $f_{12}(5)$ the value at a high emotional level:

$$edi = f_{12}(5) - f_{12}(1) \tag{1}$$

(b)　The reduction in dropout ideation attributable to effective strategies at low emotional levels can be denoted as $gsr(1)$ (good strategy reduction) in Equation (2), $gsr(1)$ being the value at a low emotional level, $f_{45}$ the emotion and dropout ideation regression line for people with good emotional regulation strategies (responses to strategy performance were 4 or 5), and $f_{45}(1)$ the regression line value at a low emotional level:

$$gsr(1) = f_{12}(1) - f_{45}(1) \tag{2}$$

(c)　The reduction in dropout ideation attributed to effective strategies at high emotional levels can be represented as $gsr(5)$ in Equation (3), $gsr(1)$ being a good strategy reduction value at a high emotional level:

$$gsr(5) = f_{12}(5) - f_{45}(5) \tag{3}$$

The final equations employed for calculating the Moderation Ratio Index (*mri*) are presented below in Equations (4) and (5):

$$mri(1) = \frac{gsr(1)}{edi} \times 100 = \frac{f_{12}(1) - f_{45}(1)}{f_{12}(5) - f_{12}(1)} \times 100 \tag{4}$$

$$mri(5) = \frac{gsr(5)}{edi} \times 100 = \frac{f_{12}(5) - f_{45}(5)}{f_{12}(5) - f_{12}(1)} \times 100 \tag{5}$$

$mri(1)$ signifies the percentage of dropout ideation increase associated with a particular emotion that effective emotional regulation strategies can mitigate at the lowest level of emotional intensity, while $mri(5)$ signifies that same percentage for the highest level of emotional intensity. Figure 1 visually illustrates the calculations mentioned above.

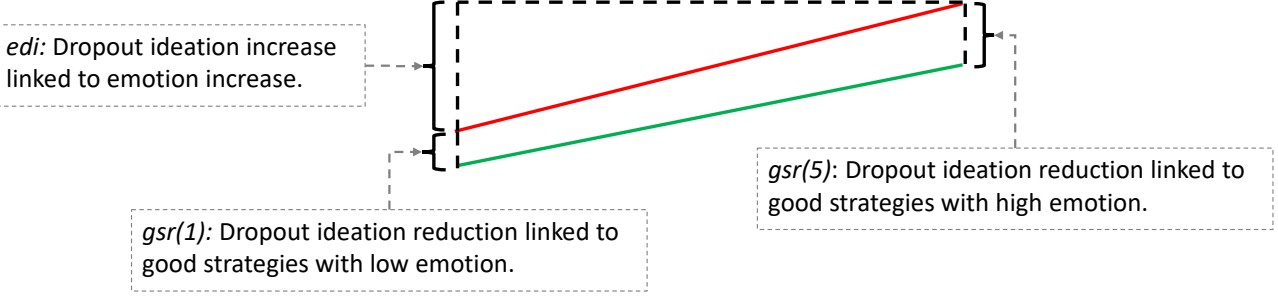

*edi:* Dropout ideation increase linked to emotion increase.

*gsr(5)*: Dropout ideation reduction linked to good strategies with high emotion.

*gsr(1):* Dropout ideation reduction linked to good strategies with low emotion.

**Legend:**

⎯⎯⎯　Regression of emotion and dropout ideation in people with poor performance emotional regulation strategies.

⎯⎯⎯　Regression of emotion and dropout ideation in people with good performance emotional regulation strategies.

**Figure 1.** Graphical explanation of the components used to calculate the Moderation Ratio Index (*mri*). The dotted lines show dropout ideation on the vertical axis and emotional intensity on the horizontal axis.

## 3. Results

To enhance comprehension of the results, they are categorized below based on the context: study, classroom, and exam. The same correlation analyses, linear regression, and scatter plots were conducted for each of these contexts.

### 3.1. Study Context

The conducted correlation analyses provide valuable insights into understanding the relationship between academic emotions and dropout ideation. As demonstrated in Table 1, all emotions exhibit statistically significant correlations with dropout ideation. Unpleasant emotions (such as anxiety, boredom, frustration, and hopelessness) display positive correlations, indicating that higher levels of these emotions in the study context are associated with increased dropout ideation. Conversely, pleasant emotions exhibit negative correlations, suggesting that higher levels of satisfaction, enjoyment, and surprise are linked to reduced dropout ideation.

**Table 1.** Correlation between academic emotions and dropout ideation in the study context.

| Academic Emotion | Dropout Ideation |
|---|---|
| Anxiety | 0.37 [0.14] * |
| Boredom | 0.18 [0.03] * |
| Frustration | 0.32 [0.10] * |
| Hopelessness | 0.39 [0.15] * |
| Satisfaction | −0.36 [0.13] * |
| Surprise | −0.19 [0.04] * |
| Enjoyment | −0.22 [0.05] * |

* All correlations (Pearson's $r$) reported are statistically significant at the $p < 0.001$ level. $R^2$ values are shown within brackets ($r[R^2]$).

The effect sizes indicate that anxiety, frustration, hopelessness, and satisfaction have the most substantial influence on dropout ideation.

Table 2 displays the connection between the efficacy of emotional regulation strategies and dropout ideation. The findings indicate that having effective strategies, both for managing pleasant and unpleasant emotions, is linked to reduced levels of dropout ideation, although the effect size is consistent but relatively small. Nevertheless, to gain a deeper understanding of the interplay between academic emotions and dropout ideation, along with the potential mediating role of emotional management strategies, it is essential to move beyond correlation values, which are constrained by their nature as measures of central tendency.

**Table 2.** Correlation between dropout ideation and the effectiveness of emotional regulation strategies in the study context.

| Effectiveness of: | Dropout Ideation |
|---|---|
| Unpleasant emotion management strategies (study context) | −0.25 [0.06] * |
| Pleasant emotion amplification strategies (study context) | −0.22 [0.05] * |

* All correlations (Pearson's $r$) reported are statistically significant at the $p < 0.001$ level. $R^2$ values are shown within brackets ($r[R^2]$).

The subsequent results present the outcomes of linear regression analyses that investigate the correlation between each emotion and dropout ideation, as well as the potential mediating influence of students' emotional management strategies. These outcomes are visually depicted using scatter plots with accompanying linear regression lines. The line and red dots on the plots represent the scores of participants who reported ineffective emotional management strategies (scores of 1 and 2 on a 1 to 5 scale), while the line and green dots indicate the scores of participants who perceive their emotional management strategies as effective (scores of 4 and 5). In the analysis of unpleasant emotions, the effectiveness of unpleasant emotional management strategies has been utilized. When examining pleasant emotions, the effectiveness of pleasant emotional management strategies was considered. When both lines in the scatter plots are parallel, it signifies that the moderation effect remains consistent across various emotional intensity levels. However, when the lines are not parallel, it indicates that the moderation effect varies at different emotional levels. The

calculation of the Moderation Rate Index (*mri*) will provide a more precise assessment of the moderation impact of emotional regulation strategies. These analyses expand our comprehension of the reported effect sizes by incorporating supplementary information.

Concerning the study context, as depicted in Figure 2, the following outcomes are evident:

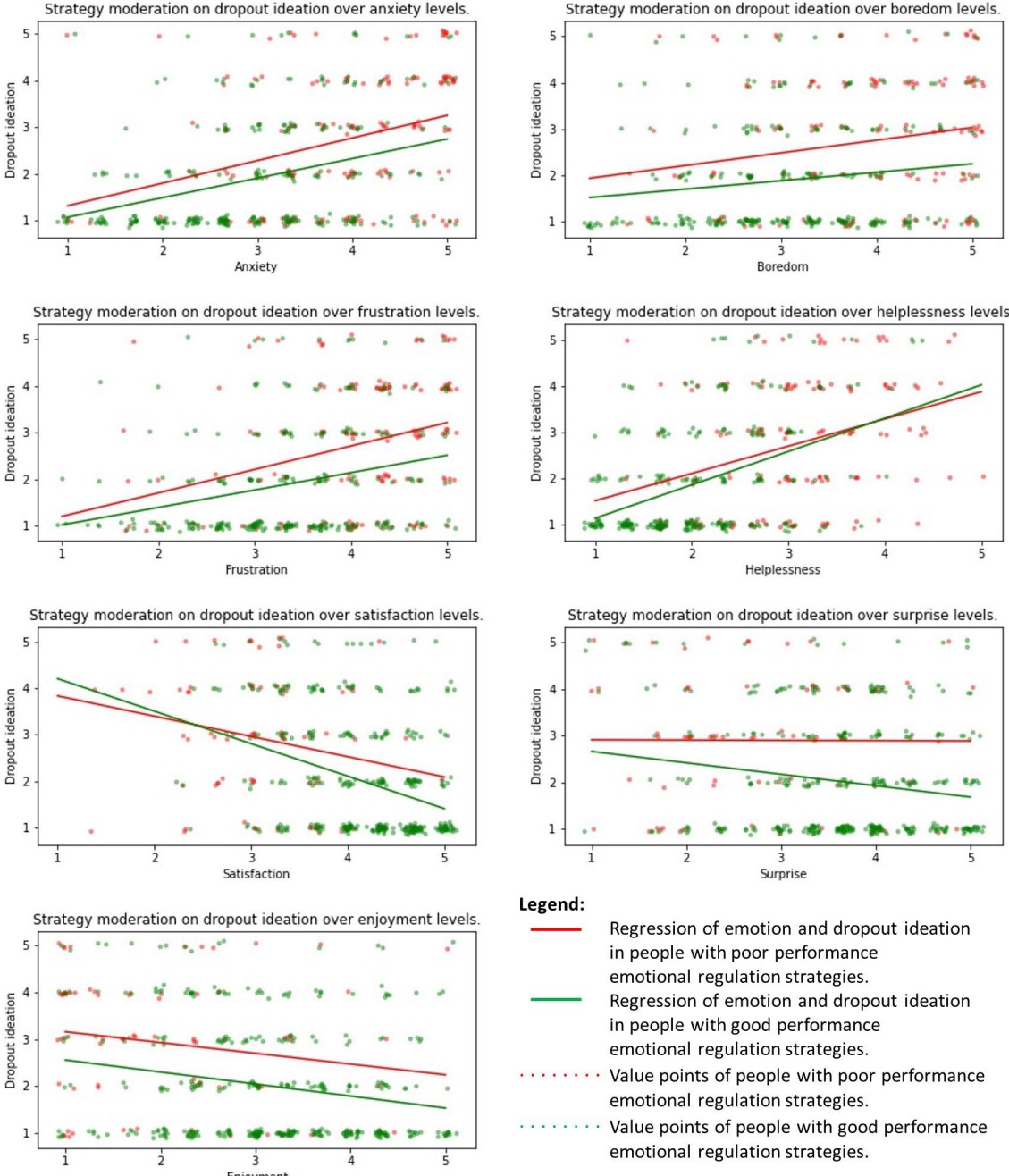

**Figure 2.** Academic emotions and emotional regulation strategies' effect on dropout ideation in the study context.

In general, it is apparent that higher levels of anxiety are linked to increased dropout ideation within the study context. The regression lines indicate that individuals who lack effective emotional management strategies tend to exhibit higher levels of dropout ideation. These lines are nearly parallel, suggesting that the mediating effect of these strategies remains consistent across varying levels of anxiety.

A closer examination of the scatter plot itself reveals that participants experiencing elevated anxiety levels, and consequently greater abandonment ideation, are predominantly those who do not possess effective emotional management strategies, as indicated by the preponderance of red dots in the upper-right section of the figure. Therefore, individuals lacking effective emotional management strategies are more inclined to experience heightened anxiety levels within the study context and, consequently, are more likely to consider abandoning their studies to a greater extent.

Concerning boredom, a comparable distribution and regression lines are evident. It could be argued, based on the shallower slope of the regression lines, that boredom exerts a somewhat lesser influence on dropout ideation compared to anxiety. However, the overall conclusion remains the same as with the previous emotion: individuals lacking effective emotional management strategies are more likely to experience elevated levels of boredom in the study context and are consequently more inclined to contemplate abandoning their studies to a greater extent.

The *mri* indicates that effective strategies mitigate the increase in dropout ideation resulting from escalating levels of boredom within a range of $mri(1) = 38.15\%$ to $mri(5) = 71.69\%$.

Once more, we encounter results like those observed for previous emotions in the case of frustration. This reaffirms the conclusion that individuals who lack effective emotional management strategies are more prone to experiencing heightened levels of frustration within the study context and are consequently more inclined to contemplate abandoning their studies to a greater extent.

Regarding hopelessness, the close intersection and proximity between the regression lines suggest the likely absence of a moderating effect of emotional management strategies. In simpler terms, it appears that regardless of the usual effectiveness of these strategies, higher levels of hopelessness are consistently linked to increased levels of dropout ideation within the study context.

Nonetheless, upon examining the scatter plot, it becomes apparent that individuals with inadequate emotional management strategies are more inclined to encounter elevated levels of hopelessness in the study context. Consequently, while emotional management strategies may not directly influence dropout ideation concerning hopelessness, they do have an impact on the level of hopelessness itself. This observation leads to the speculation that hopelessness could potentially emerge as a consequence of the absence of effective emotional regulation strategies.

Satisfaction displays a somewhat unique distribution, which slightly alters the data interpretation. The near absence of data points indicating low satisfaction directs our attention primarily to the section of the regression lines to the right of their intersection. Within this segment, it is evident that an increase in satisfaction corresponds to lower levels of dropout ideation for all participants. However, individuals with effective emotional management strategies seem to derive greater benefit from this effect. The moderation effect is not consistent but rather becomes more pronounced at higher levels of satisfaction.

In the case of surprise, a distinct moderating effect of emotional management strategies becomes evident: for individuals who lack effective strategies, surprise does not seem to have a substantial impact on their dropout ideation. However, individuals with effective strategies experience reduced levels of dropout ideation as the level of surprise increases. Consequently, students who encounter high levels of surprise while studying will derive the greatest benefit from possessing effective emotional management strategies. Once again, it is notable that the moderation effect becomes more pronounced at higher levels of surprise.

Finally, enjoyment exhibits a negative correlation with dropout ideation. Additionally, the level of dropout ideation consistently remains lower for individuals who possess effective emotional management strategies. This is evident in the *mri*, which ranges from 65.57% to 77.28%, suggesting that effective strategies significantly reduce the increase in dropout ideation associated with various levels of enjoyment.

All the regression lines have a *p*-value < 0.001, indicating that they are all statistically significant.

### 3.2. Classroom Context

As indicated in Table 3, all emotions display significant correlations with dropout ideation within the classroom context. Unpleasant emotions (anxiety, boredom, frustration, and hopelessness) demonstrate positive correlations, implying that elevated levels of these emotions in the classroom context are linked to increased dropout ideation. Conversely, higher levels of satisfaction, enjoyment, and surprise exhibit negative correlations, signifying that they are associated with lower levels of dropout ideation.

**Table 3.** Correlation between academic emotions and dropout ideation in the classroom context.

| Academic Emotion | Dropout Ideation |
|---|---|
| Anxiety | 0.42 [0.18] * |
| Boredom | 0.39 [0.15] * |
| Frustration | 0.35 [0.12] * |
| Hopelessness | 0.43 [0.18] * |
| Satisfaction | −0.42 [0.18] * |
| Surprise | −0.27 [0.07] * |
| Enjoyment | −0.43 [0.18] * |

* All correlations (Pearson's *r*) reported are statistically significant at the $p < 0.001$ level. $R^2$ values are shown within brackets ($r[R^2]$).

It is worth noting that the most robust correlations consistently appear within the classroom context. This is likely attributed to the classroom's significant role in university education and the substantial amount of time students spend in this context.

Table 4 illustrates the connection between the efficacy of emotional regulation strategies and dropout ideation, mirroring the findings in the study context. It suggests that possessing effective strategies, whether for managing pleasant or unpleasant emotions, is linked to reduced levels of dropout ideation within the classroom context.

**Table 4.** Correlation between dropout ideation and the effectiveness of emotional regulation strategies in the classroom context.

| Effectiveness of: | Dropout Ideation |
|---|---|
| Unpleasant emotion management strategies (classroom context) | −0.30 [0.09] * |
| Pleasant emotion amplification strategies (classroom context) | −0.29 [0.08] * |

* All correlations (Pearson's *r*) reported are statistically significant at the $p < 0.001$ level. $R^2$ values are shown within brackets ($r[R^2]$).

Concerning the regression analyses and scatter plots presented in Figure 3, the following results are discernible:

Anxiety, boredom, and frustration exhibit similar distributions and regression lines in the classroom context as observed in the study context. However, the mediating effect of management strategies appears to be more pronounced here, as the regression lines are more widely separated, indicating a greater reduction in dropout ideation. This observation aligns with the data shown in Table 4, which suggest that the strongest correlations between academic emotions and dropout ideation are observed within the classroom context.

Consequently, it can be inferred that individuals with inadequate emotional management strategies are more likely to experience heightened levels of anxiety, boredom, and frustration during class and are inclined to contemplate dropping out of their studies to a greater extent. Strategies are particularly effective at higher levels of anxiety (*mri* ranging from 29.71% to 46.09%) and at lower levels of boredom (*mri*: 61.82% to 52.57%) and frustration (*mri*: 57.05% to 46.22%)

In this classroom context, effective emotional management strategies do exhibit a moderating effect on hopelessness, which differs from the study context. While higher levels of hopelessness are associated with increased levels of dropout ideation for all participants, those with better strategies are less likely to contemplate abandonment. Additionally, there

is a prevalence of individuals who lack effective emotional management strategies at the highest levels of hopelessness, mirroring the observation made in the study context.

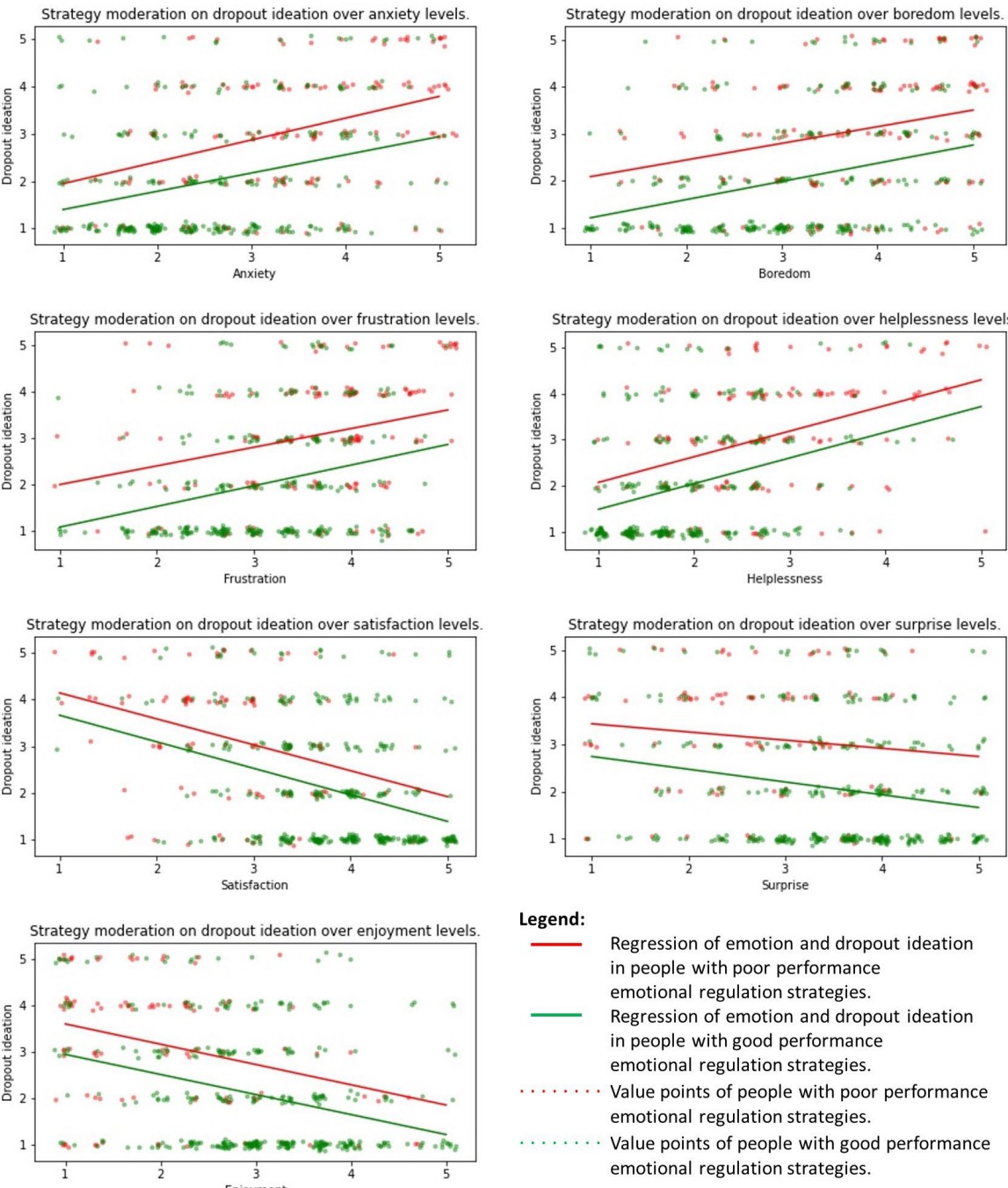

**Figure 3.** Academic emotions and emotional regulation strategies' effect on dropout ideation in the classroom context.

Within the classroom context, satisfaction demonstrates the expected effect: elevated levels of satisfaction are linked to reduced dropout ideation, and individuals with more effective emotional management strategies are less inclined to contemplate dropping out in comparison to their counterparts.

Surprise seems to have a somewhat modest dampening effect on dropout ideation, as suggested by the relatively shallow regression lines. However, the moderating effect of emotional management strategies is substantial: individuals with effective strategies exhibit

notably reduced levels of dropout ideation, especially at higher levels of surprise. Consequently, it can be inferred that strategies that enable students to enhance positive sensations arising from surprising situations have a meaningful impact on their dropout ideation.

Finally, enjoyment follows a distribution pattern similar to that observed in the study context. Elevated levels of enjoyment during classes are linked to lower levels of dropout ideation, and effective emotional management strategies contribute to diminished dropout ideation across all levels of enjoyment.

All the regression lines have a *p*-value < 0.001, indicating that they are all statistically significant.

### 3.3. Exam Context

As indicated in Table 5, all emotions display significant correlations with dropout ideation within the examination context. Unpleasant emotions (anxiety, boredom, frustration, and hopelessness) demonstrate positive correlations, signifying that elevated levels of these emotions in the examination context are associated with increased dropout ideation. However, it is important to note a notable exception, which is surprise in the examination context, showing a positive correlation. This means that higher levels of surprise in the examination context are linked to increased dropout ideation. Conversely, higher levels of satisfaction and enjoyment exhibit negative correlations, suggesting that they are associated with lower levels of dropout ideation.

**Table 5.** Correlation between academic emotions and dropout ideation in the exam context.

| Academic Emotion | Dropout Ideation |
| --- | --- |
| Anxiety | 0.23 [0.05] * |
| Boredom | 0.36 [0.13] * |
| Frustration | 0.28 [0.08] * |
| Hopelessness | 0.39 [0.15] * |
| Satisfaction | −0.35 [0.12] * |
| Surprise | 0.19 [0.04] * |
| Enjoyment | −0.18 [0.03] * |

* All correlations (Pearson's *r*) reported are statistically significant at the $p < 0.001$ level. $R^2$ values are shown within brackets ($r[R^2]$).

It is noteworthy that, in this specific examination context, surprise has altered its correlation direction with dropout ideation, with higher levels of surprise being associated with increased dropout ideation.

The results presented in Table 6 align with the findings in the other two contexts, suggesting that possessing effective emotional regulation strategies, whether for managing pleasant or unpleasant emotions, is linked to reduced levels of dropout ideation. The effect sizes, while consistent, remain relatively small.

**Table 6.** Correlation between dropout ideation and the effectiveness of emotional regulation strategies by context.

| Effectiveness of: | Dropout Ideation |
| --- | --- |
| Unpleasant emotion management strategies (exam context) | −0.28 [0.08] * |
| Pleasant emotion amplification strategies (exam context) | −0.25 [0.06] * |

* All correlations (Pearson's *r*) reported are statistically significant at the $p < 0.001$ level. $R^2$ values are shown within brackets ($r[R^2]$).

Regarding the regression analyses and scatter plots depicted in Figure 4, the following results are observed:

Before delving into the analysis of any specific emotion, it is essential to highlight a consistent pattern observed in the examination context. Divergent regression lines are evident across all emotions, except for satisfaction. This observation suggests two key points: there is a distinct moderating effect of emotional management strategies, and this effect is

notably pronounced at emotional levels corresponding to points of maximum divergence. Consequently, effective emotional management strategies appear to be particularly advantageous in the examination context for individuals who experience intense unpleasant emotions. This context enriches effect size measurements by providing additional insights into the effects of emotional regulation strategies at various emotional levels.

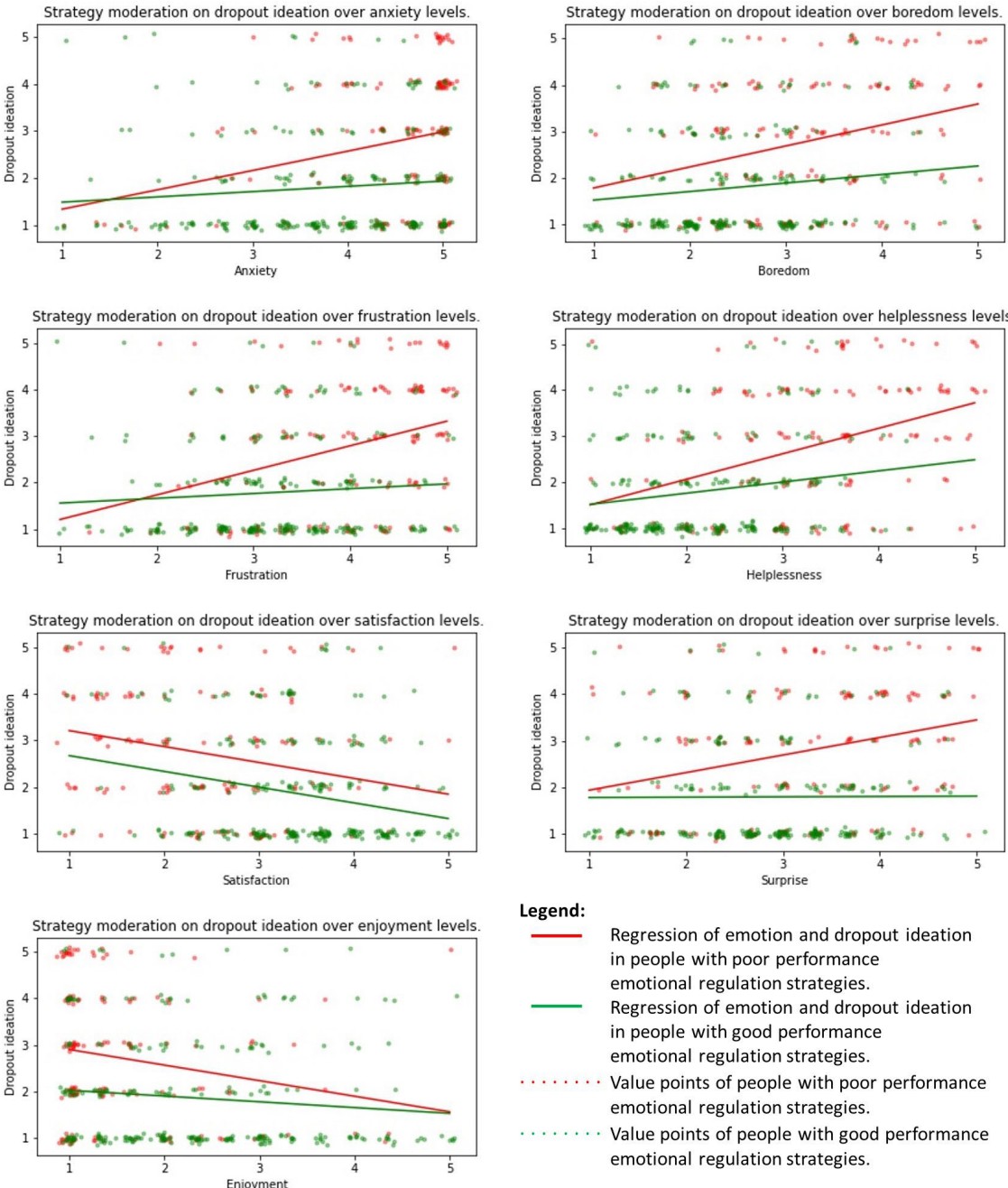

**Figure 4.** Academic emotions and emotional regulation strategies' effect on dropout ideation in the exam context.

Regarding anxiety in the examination context, there appears to be a limited relationship between anxiety and dropout ideation for individuals equipped with effective emotional management strategies. However, for those who lack such strategies, higher levels of anxiety are associated with increased dropout ideation. Consistent with observations in other contexts, the scatter plot also reveals a predominance of red data points in the upper-right quadrant. Consequently, it can be inferred that individuals without effective emotional

management strategies are more likely to experience heightened levels of anxiety related to exams and are more inclined to contemplate abandoning their studies when anxiety is elevated. The *mri* further underscores the critical role of these strategies for individuals grappling with high levels of exam-related anxiety, with an *mri* range of 9.06% to 63.71%.

Boredom, frustration, and hopelessness in the examination context display analogous distributions and regression lines to anxiety, leading to equivalent conclusions. Individuals who lack effective emotional management strategies are more prone to experience heightened levels of boredom, frustration, and hopelessness during exams and are more likely to contemplate abandoning their studies as these emotions intensify. This further underscores the significance of these strategies for individuals grappling with elevated levels of unpleasant emotions during exams.

In the examination context, satisfaction is the only emotion that does not display divergent regression lines and behaves similarly to the classroom context, albeit with a less pronounced effect. Higher levels of satisfaction correspond to lower levels of dropout ideation, and individuals with better emotional management strategies consistently exhibit lower levels of abandonment ideation. The scatter plot also illustrates that individuals lacking effective emotional regulation strategies are more likely to experience reduced levels of satisfaction.

Surprise during exams, despite being categorized as a pleasant emotion, follows a similar pattern to other unpleasant emotions in the examination context. In this instance, elevated levels of surprise are significantly linked to higher levels of dropout ideation for individuals who lack effective emotional regulation strategies (*mri*: 10.45% to 108.52%). This underscores the impact of surprise on dropout ideation in the absence of effective emotional management strategies in the examination context.

Finally, it is noticeable that, for individuals who lack effective emotional regulation strategies, lower levels of enjoyment during exams are linked to substantially higher levels of dropout ideation compared to their counterparts. As enjoyment increases in this context, the difference in dropout ideation gradually diminishes until it disappears. The scatter plot further highlights that individuals without effective emotional regulation strategies are more inclined to experience reduced levels of enjoyment during exams.

All the regression lines have a $p$-value < 0.001, indicating that they are all statistically significant.

## 4. Discussion

As previously articulated, the utilization of regression analyses, scatter plots, and *mri* calculations has facilitated a more profound elucidation of the amassed data. These analytical tools have enabled us to visually represent and gain enhanced insights into the moderating impact of emotional regulation strategies on the reduction in dropout ideation, as signified by the parallel or divergent trajectories that denote consistent or variable effects contingent upon the emotional intensity.

In the study context, four predominant emotions emerged with notable significance in the realm of correlational data: anxiety, frustration, hopelessness, and satisfaction. These emotions appear to interrelate in a manner consistent with the following hypothetical progression: decreased levels of satisfaction correspond to heightened levels of frustration within situations characterized by demanding achievement requirements. Consequently, inadequate management of frustration may precipitate a gradual erosion of perceived control over academic performance, thereby fostering elevated levels of anxiety. Ultimately, when perceived control reaches its lowest point, this amalgamation of factors can precipitate a state of hopelessness (especially in the absence of effective emotional regulation strategies), which consistently exerts the most pronounced effect on dropout ideation across all contexts.

Regarding emotional regulation strategies, the effect sizes observed were modest; however, several *mri* percentages show substantial reductions in dropout ideation. Consequently, although the overall average effect of regulation strategies on dropout ideation may appear relatively modest, these strategies exhibit noteworthy efficacy in mitigating

specific emotions and at particular intensity levels. Notably, in the study context, this moderating effect is most salient with respect to boredom and enjoyment.

Within the classroom setting, all emotions, except surprise, exhibited correlations with dropout ideation ranging between 0.35 and 0.43 for unpleasant emotions, and −0.42 to −0.43 for pleasant emotions. This context revealed the most robust correlation values, a phenomenon that is arguably reasonable given the substantial temporal and cognitive investment associated with classroom activities. The findings derived from this study underscore the profound significance of classroom experiences for students. Furthermore, these results suggest that positive teacher–student relationships are correlated with diminished dropout ideation, corroborating previous research linking positive teacher support with students' positive academic emotions [37]. However, it is imperative to acknowledge that data collection transpired well in advance of the examination period. Consequently, it is plausible that as the examination period approaches, the examination context may assume heightened prominence.

The effect size analysis indicates a limited impact of regulation strategies on dropout ideation, yet the *mri* percentages show significant reductions in dropout ideation. Notably, effective strategies primarily attenuate the escalation of dropout ideation associated with emotions such as anxiety, boredom, frustration, and enjoyment.

In the examination context, boredom, hopelessness, and satisfaction exhibited the strongest correlations with dropout ideation. The direct and indirect relationships observed for hopelessness and satisfaction are consistent with our prior expectations, where heightened hopelessness is anticipated to coincide with reduced satisfaction, both contributing to heightened dropout ideation. Conversely, the correlation identified with boredom could be construed as indicative of students experiencing reduced engagement and motivation, possibly attributable to hopelessness, and potentially fostering dropout ideation.

The effect size analysis revealed a limited impact of regulation strategies on dropout ideation, but the *mri* percentages exhibited substantial reductions in dropout ideation. This underscores the efficacy of these strategies in managing unforeseen challenges during examinations.

In summary, academic emotions emerged as a significant factor influencing dropout ideation. These findings are consistent with prior research associating emotions with academic performance [18,20–22]. Furthermore, our study revealed that emotional regulation strategies exerted a moderating role in the regulation of emotions and the reduction in dropout ideation. This finding lends further support to the limited body of scientific literature addressing this nexus [31]. It is worth noting that a study not explicitly focused on dropout ideation identified relationships between adaptive emotional regulation strategies and subjective well-being [38], a factor expected to exert an influence on student retention [23].

This study is noteworthy for its contribution to a field that has predominantly concentrated on test anxiety by introducing a broader spectrum of examination. These findings should stimulate both future research endeavors and current educational practitioners to contemplate the ramifications of diverse academic emotions on students' persistence in university. As indicated, hopelessness consistently exerts the most profound positive influence on dropout ideation, while both satisfaction and enjoyment play pivotal roles in its reduction. Therefore, even though direct control over individuals' emotions may be elusive, universities can tailor situations and tools to foster and promote specific emotions, thereby mitigating dropout ideation and potentially yielding other benefits associated with an enhanced emotional experience.

With regard to emotional regulation strategies, they exhibit substantial promise in their capacity to temper the surge in dropout ideation across all contexts and for nearly all emotions. Consequently, this study underscores the imperative for students to cultivate proficient emotional regulation strategies. In broader terms, this aligns with numerous prior studies, advocating for the inclusion of emotional intelligence, which comprehensively encompasses emotional regulation, as a fundamental component of curricula at all

educational levels, given its influence on myriad aspects of life, both within and outside the classroom [27,39,40].

This study grappled with two principal limitations. The first pertains to the uneven distribution of participants across various faculties of the Complutense University of Madrid. Certain faculties exhibited notably higher levels of participation than others, thereby skewing gender ratios (with the more engaged faculties predominantly comprising female students). Beyond gender imbalances, our study fails to offer a truly representative sample of the entire student population, as the participation of students from engineering, mathematics, and natural sciences was conspicuously limited, while psychology, medicine, and social sciences students predominated.

The second limitation is intrinsically tied to the field of educational psychology and pertains to the utilization of self-reporting measures for data collection. While self-reporting measures constitute the most widely employed instruments in the field, it is incumbent upon us to acknowledge that the gathered data are inevitably subject to bias stemming from participants' perceptions, necessitating a cautious interpretation of the results.

Future research in this domain will unfold in two principal directions. Firstly, further analyses will be conducted utilizing the comprehensive dataset at our disposal, including the application of hierarchical regression analyses. Secondly, we encourage fellow researchers to delve deeper into the domain of emotional regulation within university settings, exploring its moderating effects on diverse variables, investigating the relationship between emotional experiences and actual dropout rates (in lieu of our ideation proxy), and scrutinizing the multifaceted impacts of emotional intelligence in educational contexts. Another pertinent avenue of investigation entails strategies for fostering positive emotional experiences and developing emotional regulation skills, particularly within university settings, a realm that has thus far been less explored compared to earlier educational stages.

**Author Contributions:** Conceptualization, D.E., J.A., M.I.-S., I.S.-S.-J. and J.F.; Data curation, D.E. and J.A.; Formal analysis, D.E. and J.A.; Funding acquisition, J.A. and M.I.-S.; Investigation, J.A. and M.I.-S.; Methodology, D.E., J.A., M.I.-S., I.S.-S.-J. and J.F.; Project administration, D.E., J.A., M.I.-S. and J.F.; Resources, J.A. and M.I.-S.; Software, J.A.; Supervision, D.E., J.A., M.I.-S. and J.F.; Validation, D.E., J.A., M.I.-S., I.S.-S.-J. and J.F.; Visualization, D.E. and J.A.; Writing—original draft, D.E., J.A., M.I.-S. and I.S.-S.-J.; Writing—review & editing, D.E., J.A., M.I.-S. and I.S.-S.-J. All authors have read and agreed to the published version of the manuscript.

**Funding:** Research was funded by the Complutense University of Madrid's Student's Observatory, project number 48.

**Institutional Review Board Statement:** Reviewed and approved by the Observatorio del Estudiante de la Universidad Complutense de Madrid [Student Observatory of the Complutense University of Madrid], project nº 48.

**Informed Consent Statement:** Informed consent was obtained from all subjects involved in the study.

**Data Availability Statement:** The data presented in this study are available on request from the authors and subject to participants' explicit authorization. The data are not publicly available due to privacy.

**Conflicts of Interest:** The authors declare no conflict of interest.

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
