# Peer review of "Academic Emotions and Regulation Strategies: Interaction with Higher Education Dropout Ideation"

_education, doi:10.3390/educsci13111152_

Round 1
Reviewer 1 Report
Comments and Suggestions for Authors
Review
Introduction-
1. Line 19- ICT – write full meaning.
2. Figure 1 – no Y axis legend. Why not prepare the figure yourself or skip it? It’s not that important to your paper subject, describing these results is enough.
3. Page 3 lines 93-95. It seems that the authors should further describe paper 20 and present its results, as they are specifically relevant to their subject.
4. Page 3 lines 103-111. Most important - The tool of the questionnaire should be described in more detail. What is the purpose of the questionnaire? On which population it was previously described? What is the Cronbach Alpha of the questionnaire? What previous studies that used this questionnaire found regarding academic dropout and regarding other domains? Why were these traits chosen? What were the criteria for choosing it? How had it related to emotion and emotion regulation?
5. Page 3 line 114 – it the introduction the authors mentioned colleges students, and, in the method, they mentioned universities students. Is this the same population in both institutions? Are the dropout rates the same in both institutions?
6. The study classroom and exam context should be described in more detail. What are those factors?
Method-
7. In the introduction the authors mention that male dropout more than females. However, their participants are mostly female. Please explain how participants were recruited, and whether gender was a factor in analysis.
8. All the formulas in chapter 2.4 are not clear. In figure 2 what are the red line and the green line?
Results-
9. Table 1 – add size effect.
10. Figures 3 and 4 – the presented graphs are not in good quality and seems blurry. Can you add graphs with better quality? The meaning of the red and green lines should be described in the figure legend.
11. Add a table with all regression data (F, t, p, beta). Add all factors in the same table.
12. The authors should consider hierarchical regression, so the contribution of each factor could be understood better. Or to analyze together all positive vs. all negative emotions on dropout. What has the biggest influence? Positive or negative emotions?
Discussion-
13. The first part of the discussion (lines 428 to 489) repeats the main results but does not discuss them. The summary of the results should be much shorter.
14. The two last paragraphs should describe in more detail the relation of the results to previous studies, limitation of the study and future studies.
Comments on the Quality of English LanguageI don't have specific comments on the Engllish Language.
Reviewer 2 Report
Comments and Suggestions for Authors
The authors should ask for the help of a native English-speaking proofreader because there are some linguistic mistakes that should be fixed. The title needs further thought - shortened and more accurate.
The Abstract in its sub-sections needs re-organization and it does not adequately summarise the gist of the study.
Also, the writing style of the manuscript is not overall academic and formal. The article is proposed to be supplemented with a flowchart illustrating the research technique. A review of the literature is insufficient. It is critical to include some recent work (2018–2020) in the literature review. A literature review should be added in order to illustrate the central topic in a more detailed way. Some further explanations and interpretations are required for the results.
It is recommended to include a well-organized discussion of the findings, strengths, and limitations of the present project with additional explanation/details and a conclusion with future work.
I think the submission holds promise, but comprehensive editing is required.
Comments on the Quality of English LanguageThe authors should ask for the help of a native English-speaking proofreader because there are some linguistic mistakes that should be fixed.
